# Distance and grid-like codes support the navigation of abstract social space in the human brain

Zilu Liang[1], Simeng Wu[1], Jie Wu[1], Wen-Xu Wang[2], Shaozheng Qin[1]*, Chao Liu[1]*

[1]State Key Laboratory of Cognitive Neuroscience and Learning & IDG/McGovern Institute for Brain Research, Beijing Normal University, Beijing, China; [2]School of Systems Science, Beijing Normal University, Beijing, China

*For correspondence:
szqin@bnu.edu.cn (SQ);
liuchao@bnu.edu.cn (CL)

**Competing interest:** The authors declare that no competing interests exist.

**Abstract** People form impressions about others during daily social encounters and infer personality traits from others' behaviors. Such trait inference is thought to rely on two universal dimensions: competence and warmth. These two dimensions can be used to construct a 'social cognitive map' organizing massive information obtained from social encounters efficiently. Originating from spatial cognition, the neural codes supporting the representation and navigation of spatial cognitive maps have been widely studied. Recent studies suggest similar neural mechanism subserves the map-like architecture in social cognition as well. Here we investigated how spatial codes operate beyond the physical environment and support the representation and navigation of social cognitive map. We designed a social value space defined by two dimensions of competence and warmth. Behaviorally, participants were able to navigate to a learned location from random starting locations in this abstract social space. At the neural level, we identified the representation of distance in the precuneus, fusiform gyrus, and middle occipital gyrus. We also found partial evidence of grid-like representation patterns in the medial prefrontal cortex and entorhinal cortex. Moreover, the intensity of grid-like response scaled with the performance of navigating in social space and social avoidance trait scores. Our findings suggest a neurocognitive mechanism by which social information can be organized into a structured representation, namely cognitive map and its relevance to social well-being.

## eLife assessment

This study tackles a significant question: Does the brain apply spatial navigation systems to evaluate decision options in conceptual social spaces? The investigation is **useful** as it seeks to address this intriguing hypothesis. The findings offer partial support: a **solid** analysis revealed characteristic grid-like patterns associated with decision-making directions. However, it remains uncertain whether these effects are genuinely due to navigating a conceptual social space or potentially confounded by changes in visual stimuli. The experimental design may not be capable of definitively resolving this issue.

## Introduction

Countless daily social encounters pose people with the need to organize information about social encounters efficiently. Though people may ascribe a variety of traits to others, these inferences have been identified to rely on a few universal dimensions (*Fiske et al., 2007*; *Stolier et al., 2020*). The most widely recognized dimensions are warmth, which concerns others' intentions, and competence, which concerns others' capability and possession of resources (*Cuddy, 2008*). These universal

dimensions can be used to construct a structured representation organizing impressions of other people or other social information alike, just like how one would specify a location in a plane using orthogonal dimensions for spatial navigation. Analogous to Tolman's original concept (*Tolman, 1948*), here we termed this structured representation of social information about other people as 'social cognitive map', which may form the basis for social cognition. Having such a cognitive map for social perception is crucial. It helps us organize various experiences, track updates, and guide novel inferences efficiently (*Son et al., 2021*). Despite its importance in navigating the social world, its neural underpinning remains underinvestigated.

The resemblance between social and spatial cognitive maps has led to the proposition that social cognitive map is also supported by similar mechanisms that map physical space (*Schafer and Schiller, 2018*). In the domain of spatial navigation, it has been demonstrated that entorhinal grid cell activity supports representational coding of cognitive map. Direct recordings in rodents (*Hafting et al., 2005*) and humans *Jacobs et al., 2013* have found that the firing locations of grid cells form periodic triangular that covers the entire available arena during spatial navigation task. In this sense, grid cells form the basic units of a neural map of the spatial environment. One of the prominent properties of grid cells is their consistent grid orientation (i.e., the orientation of the grid relative to the environment) shared by neighboring grid cells (*Hafting et al., 2005*; *Sargolini et al., 2006*). Moreover, the overlapping population formed by grid cells and head direction cells as well as conjunctive grid × head-direction cells in deeper layers of the entorhinal cortex (EC) enables directional as well as positional tuning of the firing rate (*Sargolini et al., 2006*). It has been proposed that this conjunctive representation at a population level can yield a directionally modulated firing pattern with hexagonal periodicity in which population activity is enhanced when an agent's moving direction is aligned to the grid axes (*Kriegeskorte and Storrs, 2016*; *Kunz et al., 2019*). Human studies using noninvasive neuroimaging techniques rely on this directional preference of populational conjunctive grid × head-direction cell activity to detect evidence of grid-like code in functional MRI (fMRI) BOLD signals during spatial navigation (*Doeller et al., 2010*). Follow-up studies have reported entorhinal grid code during virtual as well as imagined navigation (*Horner et al., 2016*), and even mental simulation (*Bellmund et al., 2016*). More importantly, pioneering studies showed that the function of grid-like code goes beyond the scope of spatial navigation. Grid-like codes in EC and prefrontal cortex (PFC) have been observed during nonspatial navigation such as perceptual (*Aronov et al., 2017*; *Bao et al., 2019*; *Killian et al., 2012*; *Nau et al., 2018*), conceptual (*Constantinescu et al., 2016*), semantic space (*Viganò and Piazza, 2020*; *Viganò et al., 2021*), and, more recently, discrete social hierarchy (*Park et al., 2021*). Evidence from the above studies converges into a domain-general role of grid-like codes in organizing cognitive maps of various types, including spatial and nonspatial domains (*Bellmund et al., 2018*).

In this study, we tested the above hypothesis that the social cognitive map is supported by a grid cell-like coding mechanism observed in spatial navigation. First, leveraging the theoretical framework of the stereotype content model, we set up an abstract two-dimensional social value map composed of competence and trustworthiness dimensions of social perception. Next, we adapted paradigms used in human fMRI grid-code study to test the hypothesis that grid-like codes and distance codes resembling those underpinning spatial navigation are involved in navigation in this abstract social space. Finally, to complete the argument that neural codes for social navigation subserve social cognition, we explored the link between neural codes during social navigation and social skills measured by navigational performance and social trait scores.

## Results

To look for neural underpinnings of navigation in an abstract social space, we adapted a set of tasks from previous studies illustrating the relevance of grid-like code in navigating abstract concept space (*Constantinescu et al., 2016*). Participants received intensive training on navigating in this abstract social space with precision. They completed a learning session, a review session during behavioral training on the first day, and a scanning session on the second day (*Figure 1—figure supplement 1F*).

We designed the social value map, an abstract social space structuring one's social perception of other people's social values. It was defined by two ecologically important dimensions: competence and trustworthiness. A visualization with two adjacent bars enclosed by a square box was designed to represent the location on this social value map, with the height of each bar representing the value of each dimension. We operationally defined these two dimensions under the framework of an

investment game, in which the trustee's social value is determined by the ability to earn a profit (i.e., profit rate as competence) and the proportion of earnings returned to the investor (i.e., return rate as trustworthiness). The range of competence and trustworthiness was limited to between 0 and 1 to make sure that they are realistic (i.e., it is unlikely that trustees will return you more than they have gained), and more importantly, orthogonal and share the same metric system. Participants played the role of an investor in this investment task (*Figure 1—figure supplement 1A*) at the very beginning of the experiment to develop a concrete understanding of the visual analog and the quantitative meaning of these two dimensions.

Six avatars with different levels of competence and trustworthiness (*Figure 1A*), analogous to landmarks in the physical environment, were placed on the social value map of size 1 unit × 1 unit in accordance with the range of competence and trustworthiness. The current spatial arrangement aimed to make avatars distinguishable on both dimensions while spreading widely across the whole space. Specifically, each participant's set of avatars' coordinates was sampled within circles of 1/30-unit radius around center coordinates (*Figure 1B*). In the experiment, avatars were represented by passport-style photos of the faces of volunteers taken against a plain blue background. Photographs were matched on competence and trustworthiness ratings based on data from an independent online sample. Additionally, scanned participants completed ratings on the selected faces before and after the experiment to reassure that there was no pre-existing bias in social perception of the avatars and to test whether they updated social perception according to learned characteristics of the avatars.

In the training, participants explored the social space by morphing the bars with different competence:trustworthiness ratio using the nonspatial controller to look for the six avatars (*Figure 1—figure supplement 1C*). Morphing the bars resembles making a transition in the social space in that it was done in two steps: (1) deciding a trajectory to follow by changing the competence:trustworthiness ratio using the nonspatial controller, and (2) how further along to follow that trajectory. No prior information about the characteristics of the avatars was provided to the participants. Thus, participants could only explore social space as if they were looking for landmarks in a newly introduced physical environment. An avatar would pop out when the visualization matched his/her characteristics. Specifically, the avatar would pop out when participants' current location fell within a 0.01-unit radius of the correct location on the social value map. In this way, participants not only learned the characteristics of each avatar but also became familiar with the whole space even though this map-like structure was never revealed to them.

After participants were fully acquainted with the avatars and the social space, a set of memory tasks were used to test whether participants learned the avatars, and, more importantly, if they formed an internal representation of the social value map, even though the map-like structure was never directly revealed to them. Participants never received feedback on these memory test tasks, so they had to rely on knowledge learned in the explore task. The first one was the collect task (*Figure 1—figure supplement 1D*), which required participants to navigate to the location of the avatars from random starting locations in the abstract space. In each trial, they were instructed to morph the box on the left to match the characteristic of the avatar shown on the right. Specifically, we asked participants to make as few transitions as possible. Two performance indices were calculated: (1) the deviation of participants' first transition from the ideal trajectory, and (2) the Euclidean distance between the location corresponding to participants' morphed bars and the avatar's actual location in the abstract space. The second one was the recall task, which required participants to mentally traverse the abstract social space according to a half-seen trajectory and identify the destination that follows (*Figure 1D*, *Figure 1—animation 1*). In each trial, participants were first shown the two bars morphing according to a predefined competence:trustworthiness ratio for 1 s. The bars then stopped morphing, and participants were instructed to imagine the bars keep morphing according to the same ratio, at the same speed, and for the same amount of time. After this, participants had to choose which of the three given options matched the bars after imagination. Essentially, each trial is a trajectory vector defined by moving direction (i.e., morphing ratio) and traveled distance on the social value map. We specifically sample the trajectories from a uniform distribution (*Figure 1—figure supplement 2*). We investigated the neural representation of the trajectories while participants performed this task. The third one was the compare task (*Figure 1—figure supplement 1E*), which required participants to compare different pairs of avatars on one given dimension (competence and trustworthiness) or by a fair combination of both dimensions (willingness to cooperate). Assuming distance is a measure of

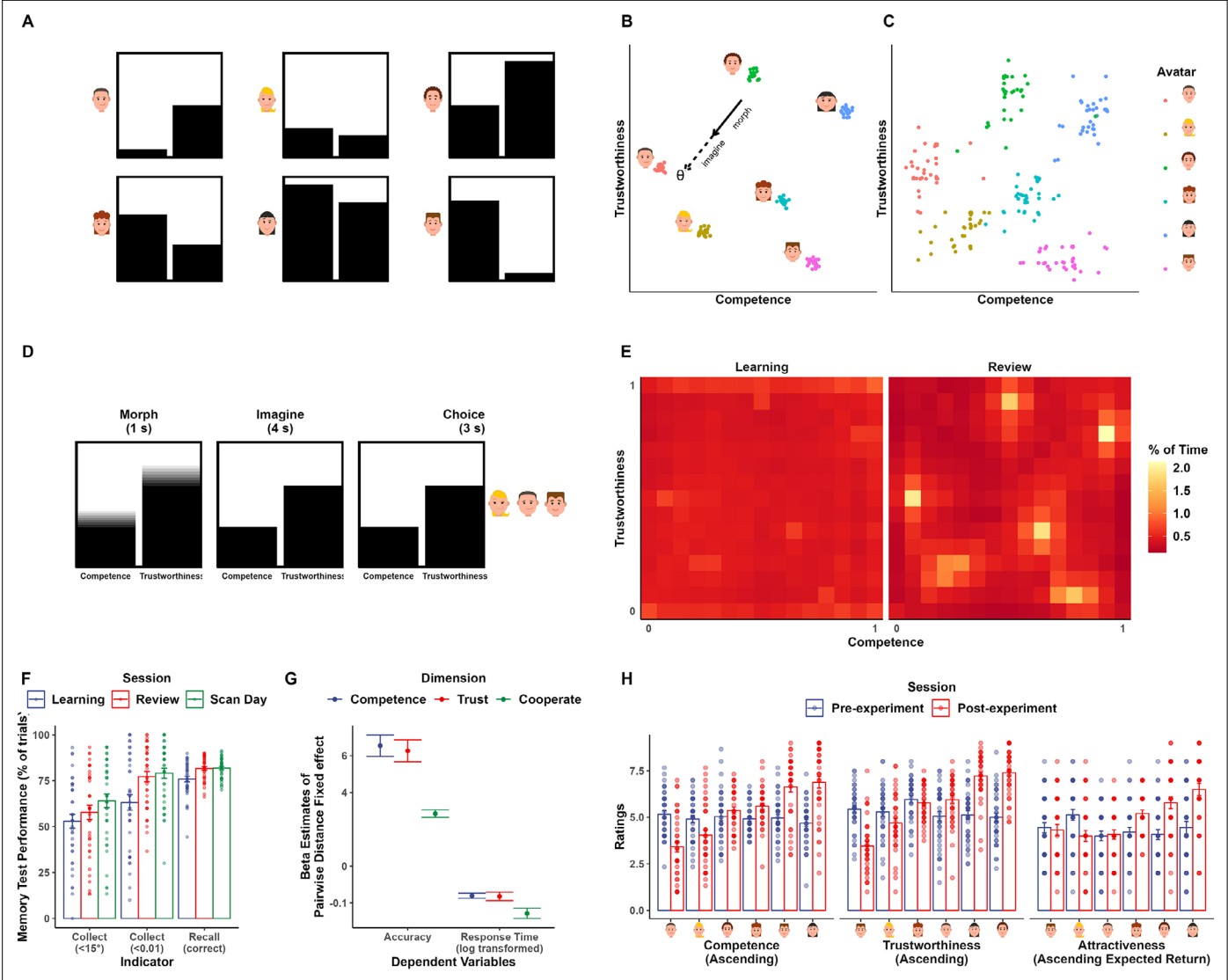

**Figure 1.** Experimental design and behavioral performance. (**A**) Visual analogs illustrating each avatar's social values. The height of the left bars signals values on the competence dimension, while the height of the right bars signals trustworthiness. Labels for each of the two bars are omitted for illustration purposes. (**B**) Corresponding layout of the social value map and example trajectory in the recall task. Scattered dots indicate the actual sets of positions generated for participants. (**C**) Positions of the avatars indicated by participants at the end of the whole experiment. (**D**) Trial timeline of the recall task. Participants first watched the bars morphing for 1 s according to a predefined competence-trustworthiness ratio (i.e., the first half of a trajectory in **B**). After the bars stopped morphing, they were instructed to imagine the bars continue to morph according to the same ratio, at the same speed, and for the same amount of time (i.e., the second half of a trajectory in **B**). (**E**) Color-coded trajectory map of the explore task during the learning and review session. Color indicates the percentage of time spent in each pixel of the social value map during the explore task in the learning (left panel) and review (right panel) session. A pattern emerged that participants spent more time at avatars and less time at the edge during review. (**F**) Memory test performance of the collect and recall task in learning, review, and scan-day sessions. Collect (<15°): percentage of trials where the deviation of first transition from ideal trajectory was smaller than 15°; Collect (<0.01): percentage of trials where the distance between response location and correct location was smaller than 0.01 units; Recall (correct): percentage of trials where the response was correct. (**G**) The distance effect in comparison task across the review and scan-day session. Colored dots indicate the estimates of distance effect regressor while error bars indicate standard error of the estimates. (**H**) Ratings became predictable from the avatars' social value after the experiment. Icons of avatars are for illustration and retrieved from https://pixabay.com/vectors/avatar-flat-modern-minimal-5261900/, https://pixabay.com/vectors/avatar-flat-modern-minimal-5261896/.

The online version of this article includes the following video and figure supplement(s) for figure 1:

**Figure supplement 1.** Behavioral training tasks.

**Figure supplement 2.** Distribution of trajectory direction (theta) in the recall task in the scanner.

**Figure supplement 3.** Distribution of trajectory length (travelled distance) in the recall task in the scanner.

**Figure 1—animation 1.** Recall task morph stage.

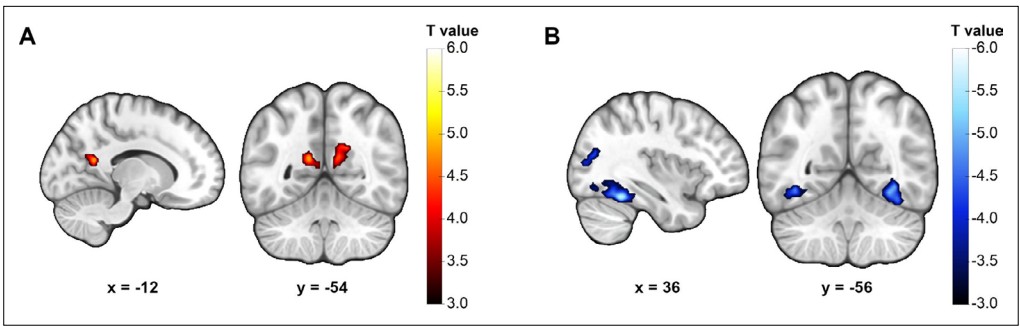

**Figure 2.** Neural representation of Euclidean distance on the social value map. (**A**) Activity in the bilateral precuneus positively correlated with traveled Euclidean distance. (**B**) Activity in bilateral fusiform and the right middle occipital gyrus negatively correlated with traveled Euclidean distance. Display threshold: cluster-defining threshold p<0.001.

similarity between two avatars, the closer two avatars are, the more similar they are, hence distinguishing them will be harder, which results in less accurate response and longer reaction time (similar to the inferential distance effect in ordered relationships; *Potts, 1974*). If participants had map-like representations, then their response in this task would show such a correlation pattern with distance in the 2D social space. The final one was a map task at the end of the experiment in which participants were informed about the map-like structure and were asked to indicate the location of each avatar on an empty social value map using a mouse click.

## Participants construct a social value map after associative learning of avatars and corresponding characteristics

Participants were successful in reconstructing the designed layout of avatars in a map task at the end of the experiment (*Figure 1C*). Linear mixed effect models revealed that participants' performance in tasks significantly improved over sessions. In the explore task, it took them significantly less time to find all avatars in the review session compared to the learning session ($\beta_{session}$ = –67.589, $t$ = –37.604, p<0.001). They also spent much less time at the edge ($\beta$ = –0.104, $t$ = –9.070, p<0.001) and more time at avatars ($\beta_{session}$ = 0.023, $t$ = 7.367, p<0.001) during exploration in the review session compared to the learning session (*Figure 1E*). Likewise, there was evident improvement in collect and recall task as training progressed (*Figure 1F*, percentage of first transition deviation < 15° in collect task: $\beta_{session}$ = 0.050411, $t$ = 3.2187, p=0.002; percentage of distance < 0.01 in collect task: $\beta_{session}$ = 0.047, $t$ = 3.002, p=0.003; accuracy in the recall task: $\beta_{session}$ = 0.020, $t$ = 4.213, p<0.001). Furthermore, participants were able to form an implicit map-like structure and integrate both dimensions when making decisions between avatars. In the compare task, linear mixed effect models revealed a significant effect of distance between avatars in the compared dimension on response time and accuracy as predicted (*Figure 2G*, all p<0.001). Lastly, learning-induced changes in participants' ratings of avatars on all rating items (competence, trustworthiness, and attractiveness) (*Figure 2H*, *Table 1A*). There was no preference over the six presented faces when participants were naïve to the social value of different avatars (*Table 1B*, all p>0.890), but their ratings became significantly dissociable according to the learned social value at the end of the experiment (*Table 1B*, all p<0.001). These results indicated the successful formation of a 2D social map by associating each avatar with its respective social value.

## Precuneus and fusiform jointly encode Euclidean distance during social navigation

Next, we investigated the neural representations supporting the social cognitive map. To examine the neural code for distance information, we conducted whole-brain analysis with a GLM with two regressors modeling the morph stage and choice stage. Traveled Euclidean distance (i.e., the length of each trial's trajectory vector) was entered as a parametric modulator of the morph stage regressor. Even though we did not specifically control for the distribution of traveled distance, there was quite a lot trial-wise variability that allowed us to conduct this test (*Figure 1—figure supplement 3*).

**Table 1.** Related to *Figure 1H*.

**A. Linear mixed effect model for different rating items**

| | Competence | Trustworthiness | Attractiveness |
|---|---|---|---|
| (Intercept) | 5.157 *** [4.775, 5.539] | 5.614 *** [5.276, 5.951] | 4.590 *** [4.168, 5.011] |
| Post vs pre | –2.173 *** [–2.707, –1.639] | –1.998 *** [-2.469,–1.527] | –0.828 ** [–1.364, –0.292] |
| Avatar | –0.395 [–1.024, 0.234] | –0.637 * [–1.240, –0.034] | –0.281 [–0.710, 0.149] |
| avatar * (post vs pre) | 4.812 *** [3.923, 5.702] | 5.185 *** [4.333, 6.038] | 1.964 *** [1.357, 2.572] |
| N (observation) | 456 | 456 | 456 |
| N (id) | 38 | 38 | 38 |
| AIC | 1600.834 | 1589.456 | 1808.221 |
| BIC | 1625.569 | 1614.191 | 1832.956 |
| R2 (fixed) | 0.302 | 0.337 | 0.129 |
| R2 (total) | 0.313 | 0.347 | 0.223 |

**B. Follow-up analysis of interaction term in mixed effect models: simple slope of avatars in different sessions**

| Rating item | Moderator levels session | Estimate [lower CI, upper CI] | SE | t (415) | p |
|---|---|---|---|---|---|
| | Pre-experiment | –0.395 [-1.026, 0.236] | 0.321 | –1.231 | 0.89 |
| Competence | Post-experiment | 4.417 [3.786, 5.048] | 0.321 | 13.766 | <0.001 |
| | Pre-experiment | –0.637 [-1.241, 0.032] | 0.308 | –2.069 | 0.98 |
| Trustworthiness | Post-experiment | 4.549 [3.944, 5.154] | 0.308 | 14.787 | <0.001 |
| | Pre-experiment | –0.281 [-0.711, 0.150] | 0.219 | –1.281 | 0.9 |
| Attractiveness | Post-experiment | 1.684 [1.253, 2.114] | 0.219 | 7.684 | <0.001 |

Statistic results for right-sided *t*-test against zero (noninferiority).

AIC=Akaike Information Criterion ; MNI=Montreal Neurological Institute ; ACC=Anterior Cingulate Cortex ; FWE=Family Wise Error ; DMN=Default Mode Network， SAD=Social Anxiety Disorder ; BIC=Bayesian Information Criterion.

*p<0.05; **p<0.01; ***p<0.001.

This analysis revealed two marginally significant clusters in the precuneus whose activity positively correlated with traveled distance on the social value map (*Figure 2A*, *Table 2A*, right precuneus: peak coordinate [*x, y, z*] = [12, –56, 26], *t*(37) = 4.952, p=0.056 FWE-corrected; left precuneus: peak coordinate [*x, y, z*] = [-12, –54, 18], *t*(37) = 4.530, p=0.058 FWE-corrected; cluster-defining threshold p<0.001 FWE-corrected). Interestingly, we also observed negative correlation between traveled distance and

**Table 2.** Related to *Figure 2*.

Neural codes represent traveled distance on the social value map.

| | | Peak MNI coordinates | | | | Cluster | |
|---|---|---|---|---|---|---|---|
| Anatomical description | Hemisphere | x | y | z | Peak *t*-value | Size | p_FWE |
| **A. Regions positively correlated with traveled distance** | | | | | | | |
| Precuneus | R | 12 | –56 | 26 | 4.53 | 151 | 0.054 |
| Precuneus | L | –12 | –54 | 18 | 4.952 | 134 | 0.082 |
| **B. Regions negatively correlated with traveled distance** | | | | | | | |
| Fusiform | R | 36 | –48 | 20 | 4.53 | 766 | <0.001 |
| Fusiform | L | –38 | –56 | –10 | 4.952 | 234 | 0.008 |
| Middle occipital gyrus | R | 42 | –80 | 8 | 4.952 | 196 | 0.018 |

L, left; R, right.

**Table 3.** Regions showing hexagonal modulation (GLM1).

| Anatomical description | Hemisphere | Peak coordinates (MNI) | | | Peak t-value | Cluster | |
|---|---|---|---|---|---|---|---|
| | | x | y | z | | Size | p$_{FWE}$ |
| Superior parietal gyrus | R | 28 | –66 | 56 | 5.021 | 522 | <0.001 |
| Precuneus | L | -6 | –46 | 38 | 5.707 | 426 | <0.001 |
| Middle frontal gyrus | R | 42 | 24 | 34 | 5.221 | 271 | <0.001 |
| Paracentral Lobule | R | 10 | –36 | 68 | 4.835 | 120 | <0.001 |
| Middle frontal gyrus | R | 38 | 42 | 30 | 4.849 | 103 | 0.001 |
| Frontal pole | L | –26 | 50 | 0 | 4.269 | 71 | 0.014 |
| Frontal pole | R | 26 | 52 | -2 | 4.494 | 69 | 0.017 |
| Angular | L | –52 | –60 | 26 | 4.718 | 58 | 0.041 |

L, left; R, right.

activity of clusters in several regions, including bilateral fusiform gyrus (*Figure 2B*, *Table 2B*, right fusiform: peak coordinate [*x, y, z*] = [36, –48, –20], *t*(37) = 5.912, p<0.001 FWE-corrected; left fusiform: peak coordinate [*x, y, z*] = [–38, –56, –10], *t*(37) = 4.724, p=0.008 FWE-corrected) and the right middle occipital gyrus (*Figure 2B*, *Table 2B*, peak coordinate [*x, y, z*] = [34, –68, 20], *t*(37) = 4.649, p=0.018 FWE-corrected; cluster-defining threshold p<0.001).

## Grid-like activity aligned to prefrontal and entorhinal grid orientation

To look for grid-like activity, we implemented the orientation-estimation approach (*Bellmund et al., 2018*; *Doeller et al., 2010*) based on univariate analysis. First, a whole-brain quadrature filter analysis was conducted, which served as a functional localizer to identify the regions sensitive to hexagonal modulation, independent of grid orientation ('Materials and methods'). This GLM had two regressors modeling the morph stage and choice stage. For the morph stage regressor, we included two parametric modulators corresponding to the sixfold sinusoidal sixfold modulation of trajectory direction (i.e., sin6θ and cos6θ, θ is the direction). Next, we defined spherical region of interest (ROI) with a 5 mm radius centered at the peak coordinate of each significant cluster (*Table 3*) to estimate the grid orientation of each ROI. As no region in the EC survived this analysis, we additionally defined four entorhinal ROIs based on subdivisions of EC (left/right and posterior–medial/anterior–lateral, i.e., pmEC/alEC) from an anatomical mask (*Maass et al., 2015*). Finally, we conducted the leave-one-out cross-validation analysis to test for hexagonal modulation aligned to grid orientation of the ROIs for each participant, that is, the grid orientation consistency effect ('Materials and methods'). In this cross-validation approach, we estimated grid orientation using data from three runs and tested the grid orientation consistency effect in the held-out run. Grid consistency effect was tested using a GLM that classified trials into 12 bins according to its trajectory direction offset from the estimated grid orientation and built a contrast to test whether activity is stronger in aligned trials than in misaligned trials (*Figure 3A*).

Of all the grid orientations derived from the above ROIs, we only found grid consistency effect aligned to the grid orientation of the ROI in the right frontal pole (right FP) cluster (*Figure 3B*, peak coordinate [*x, y, z*] = [26, 52, –2], *t*(37) = 4.494, cluster-level p=0.017 FWE-corrected, cluster-defining threshold p<0.001) and the grid orientation of the anatomically defined right posterior–medial EC (right pmEC, *Figure 3C*). For the right FP grid orientation, whole-brain analysis revealed consistency effect in bilateral ventral medial PFC (vmPFC, *Figure 3D*, left and middle panels, peak coordinate [*x, y, z*] = [–14, 56, –12], *t*(37) = 4.245, cluster-level p=0.033 FWE-corrected, cluster-defining threshold p<0.005) and the left EC (left EC) extending to the parahippocampus (*Figure 3E*, left and middle panels, peak coordinate [*x, y, z*] = [–16, –22, –26], *t*(37) = 4.563, cluster-level p=0.024 FWE-corrected, cluster-defining threshold p<0.005). For the right pmEC grid orientation, significant consistency effect was found in two clusters in bilateral vmPFC (*Figure 3F*, left and middle panels, right rectus: peak coordinate [*x, y, z*] = [10, 34, –18], *t*(37) = 4.908, cluster-level p=0.033 FWE-corrected; left ACC: peak coordinate [*x, y, z*] = [–8, 32, 6], *t*(37) = 4.810, cluster-level p=0.009 FWE-corrected; cluster-defining

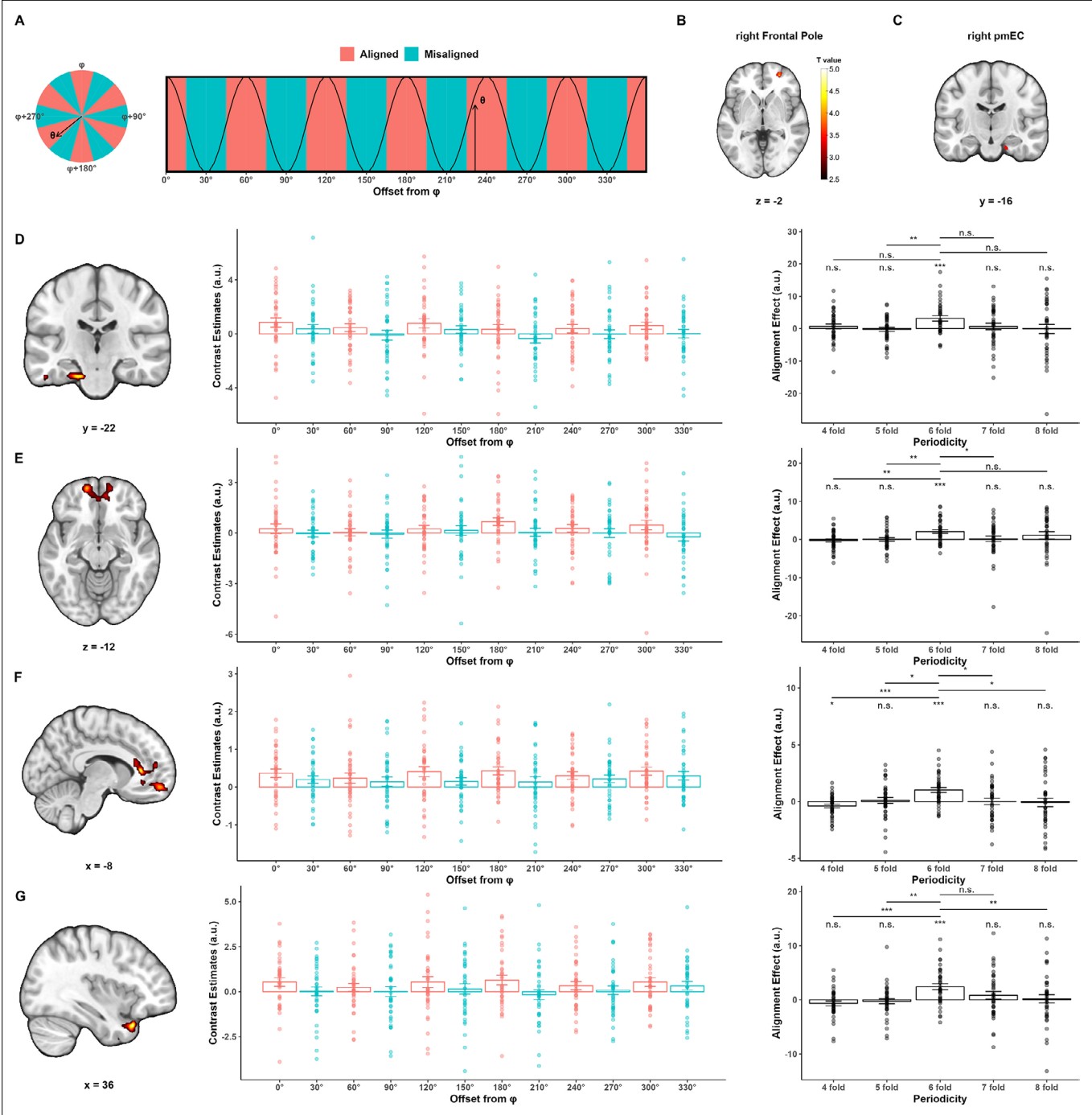

**Figure 3.** Evidence of grid-like activity aligned to putative grid orientation in the right frontal pole and the right posterior–medial entorhinal cortex. (**A**) Theoretical prediction of grid-like activity. (**B, C**) Regions of interest (ROIs) for deriving putative grid orientations: (**B**) right FP ROI from quadrature filter analysis showing sensitivity to hexagonal modulation. A 5 mm sphere was defined around the peak coordinate to compute grid angle. Display threshold: voxel-level p<0.001, cluster-level p<0.05 FWE-corrected. (**C**) Anatomically defined right pmEC ROI used to compute grid angle. (**D–G**) Grid-like activity aligned to putative grid orientations in the right FP ROI (**D, E**) and right pmEC (**F, G**) ROI, respectively. Left panels: clusters from whole-brain hexagonal consistency analysis. Color indicates *T* statistics as shown in the colorbar in (**B**). Display threshold: voxel-level p<0.005, cluster-level p<0.05 FWE-corrected. Middle panels: hexagonal consistency effects plotted as contrast estimates of the 12 trial-bin regressors extracted from corresponding cluster in the left panel; To illustrate the effect in EC in (**E**), estimates were extracted from the intersection of the suprathreshold cluster and anatomical mask of the EC. Right panels: such effects were specific to sixfold. n.s., p>0.05, *p<0.05, **p<0.01, ***p<0.001. FP, frontal pole; pmEC, EC, entorhinal cortex; posterior–medial entorhinal cortex; vmPFC, ventral medial prefrontal cortex; STP, superior temporal pole.

The online version of this article includes the following figure supplement(s) for figure 3:

*Figure 3 continued on next page*

*Figure 3 continued*

**Figure supplement 1.** Analysis pipeline of multivariate pattern analysis in entorhinal region of interest (ROI).

**Figure supplement 2.** Region of interest (ROI) analysis of univariate and multivariate grid-like code in the entorhinal cortex.

**Figure supplement 3.** Distribution of voxel-wise grid orientation of example participants (voxel-wise distribution plot of all participants can be viewed at https://doi.org/10.57760/sciencedb.08637).

**Figure supplement 4.** Distribution of grid orientation across participants.

**Figure supplement 5.** Relationship between temporal signal-to-noise ratio (tSNR) and the strength of evidence of hexagonal modulation effect in frontal pole and entorhinal regions of interest (ROIs).

threshold p<0.005) and one cluster in right superior temporal pole (right STP, *Figure 3G*, left and middle panels, peak coordinate [*x, y, z*] = [36, 20, –30], *t*(37) = 4.962, cluster-level p=0.018 FWE-corrected, cluster-defining threshold p<0.005). The above analysis did not identify grid-like pattern in EC that is aligned to the grid orientation of itself. It could be that the signal-to-noise ratio is too low in EC to pass the stringent whole-brain test. Another possibility is that a distributed coding scheme is employed by the EC. However, in the more statistically lenient ROI-based analysis, we failed to find evidence of grid-like code in the EC aligned to its own putative grid orientation either with this orientation estimation approach or with the representation similarity analysis (RSA) approach ('Materials and methods', *Figure 3—figure supplements 1 and 2*).

One important assumption underlying the univariate analysis for grid-like code in human fMRI data is that neighboring grid cells share similar grid orientation. Consistent with previous studies, we examined this assumption in our dataset by testing the distribution of putative voxel-wise grid orientations in the right FP ROI and in the right pmEC ROI for each participant ('Materials and methods'). This revealed a clustered distribution of voxel-wise grid orientations in FP ROI (*V*-test, all p<0.011, except in one participant's estimating set p=0.176; see *Figure 3—figure supplement 3A* for representative participants). Previous studies using a similar approach in fMRI have also shown that grid orientations across participants tend to distribute uniformly (*Bao et al., 2019*). This has been replicated in the right FP ROI in our dataset (*Figure 3—figure supplement 4*, top, Rayleigh's tests for nonuniformity, all p>0.782). However, in the pmEC ROI, these results are less consistent. With regard to voxel-wise orientations, 31 participants show the above clustered distribution (*V*-test, all p<0.040; see *Figure 3—figure supplement 3B* for representative participants), but there are seven participants each of whom has at least one estimating set that did not pass the statistical test (*V*-test, all p>0.068). Across participants, only grid orientations from two estimating sets conformed to the uniform distribution (*Figure 3—figure supplement 4*, bottom panel, Rayleigh's tests for nonuniformity, run 1,2,3: p=0.010, run 1,2,4: p=0.019).

For completeness, we tested the specificity of hexagonal modulation. That is, the identified alignment effect only exists with sixfold periodicity. The above orientation estimation and cross-validation procedure was repeated for each of the four controlled periodicity (i.e., fourfold, fivefold, sevenfold, and eightfold). Signals were extracted from the clusters that showed significant hexagonal consistency. One-sample *t*-tests showed that signals within these clusters were not modulated by any of the controlled periodicities, and paired *t*-test showed that sixfold periodicity in general showed greater alignment effect (*Figure 3D–G*, right panels). In sum, our univariate analysis revealed that the vmPFC and left EC showed sixfold-specific grid-like code aligned to the right FP's grid orientation and that the vmPFC and right STP showed sixfold-specific grid-like code aligned to the right pmEC's grid orientation.

## Behavioral relevance of spatial codes for the social value map

Next, we want to address whether the identified distance or grid-like code could have any behavioral relevance. Previous studies have found that hexagonal modulation effect and hexagonal consistency effect positively correlate with accuracy in the recall task in the scanner (*Constantinescu et al., 2016*). However, we did not find such a correlation in our data (*Figure 4—figure supplement 1*). Then, we explored if there is any correlation between neural indices of map-like representation and behavior performance outside the scanner as well as individual differences. The covariates we explored can be classified into two categories. The first category reflects participants' ability to make decisions based on the social value map, that is, the distance effect in the compare task. Specifically,

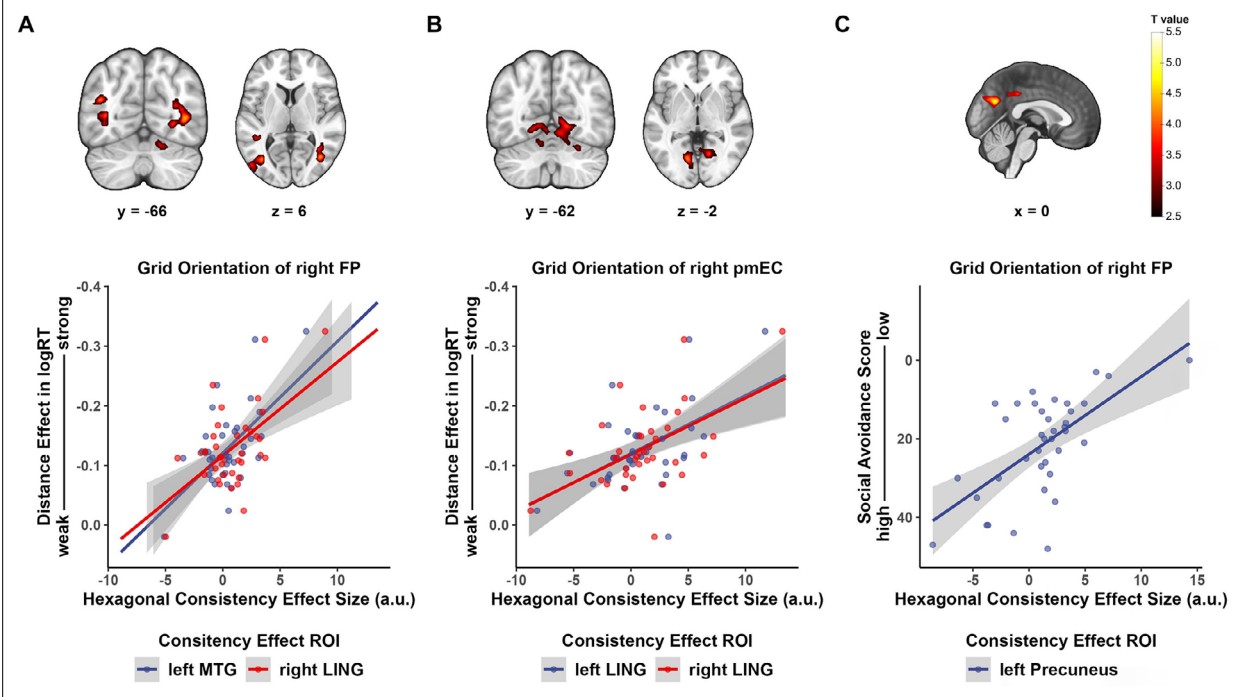

**Figure 4.** Behavioral relevance of hexagonal consistency effect. (**A, B**) Higher hexagonal consistency in temporal lobe aligned to grid orientation of (**A**) right FP region of interest (ROI) and (**B**) right pmEC significantly correlated with stronger distance effect in compare task when choosing preferred partners for cooperation. Display threshold: voxel-level p<0.005, cluster-level p<0.05 FWE-corrected. (**C**) Hexagonal consistency effect in left precuneus aligned to grid orientation of right FP ROI significantly correlated with social avoidance score. Display threshold: voxel-level p<0.001, cluster-level p<0.05 FWE-corrected. n.s., p>0.05, *p<0.05, **p<0.01, ***p<0.001. FP, frontal pole; pmEC, posterior–medial entorhinal cortex; MTG, middle temporal gyrus; LING, lingual gyrus.

The online version of this article includes the following figure supplement(s) for figure 4:

**Figure supplement 1.** No evidence of correlation between grid-like (**A**) and distance (**B**) representation and performance in the scanner.

we focused on the distance effect during the cooperation block when participants choose the avatar they are more willing to cooperate with by taking into account both dimensions. This is essentially the beta weight of the distance regressor in predicting participants' accuracy and log-transformed response time. The second category reflects participants' social trait, including social anxiety and social avoidance scores. The covariates were entered into the second-level analysis of the consistency effect GLM separately.

Whole-brain analysis revealed significant clusters in the temporal lobe whose consistency effect scaled with distance effect in response time. For grid orientation of right FP ROI (*Figure 4A*), hexagonal consistency effect in the left middle temporal gyrus and right lingual gyrus correlated with the distance effect (left middle temporal gyrus: peak coordinate [x, y, z] = [–36, –70, 8], t(37) = 4.528, cluster-level p<0.001 FWE-corrected; right lingual gyrus: peak coordinate [x, y, z] = [–38, –66, 4], t(37) = 4.369, cluster-level p<0.001 FWE-corrected; cluster-defining threshold p<0.005). For grid orientation of right pmEC ROI (*Figure 4B*), hexagonal consistency effect in bilateral lingual correlated with the distance effect (left lingual gyrus: peak coordinate [x, y, z] = [–12, –68, –2], t(37) = 4.355, cluster-level p=0.077 FWE-corrected; right lingual gyrus: peak coordinate [x, y, z] = [16, –62, –2], t(37) = 3.934, cluster-level p=0.032 FWE-corrected; cluster-defining threshold p<0.005). In general, greater distance effect in the comparison task correlated with greater hexagonal consistency in temporal lobe aligned to both prefrontal and entorhinal grid orientations. In addition, this analysis also revealed a cluster in the left precuneus/posterior cingulate cortex (left PCC, *Figure 4C*, peak coordinate [x, y, z] = [0, –64, 28], t(37) = 4.965, cluster-level p<0.001 FWE-corrected, cluster-defining threshold p<0.001). Its consistency effect aligned to the right FP grid orientation was negatively correlated with social avoidance score.

# Discussion

In this study, we investigated whether cognitive map in social cognition domain recruits the neural processing mechanism similar to its spatial counterpart. We identified distance representation in the precuneus, fusiform gyrus, and middle occipital gyrus. In addition, we demonstrated some evidence of grid-like activity in PFC and EC. Furthermore, we explored the behavioral relevance of grid-like activity in temporal pole and precuneus.

When participants were mentally traversing a predefined social value map, we found that activities in the bilateral precuneus were positively correlated with traveled distance, while activities in bilateral FFG and right MOG were negatively correlated with traveled distance. In spatial navigation literature, though intracranial electroencephalography studies have shown the relevance of hippocampal theta oscillation in encoding traveled distance, neural correlates of traveled distance at the macroscopic level using fMRI are relatively sparsely investigated (*Kunz et al., 2019*). *Patai et al., 2019* showed that retrosplenial cortex positively correlated with path distance in that greater distance is associated with increased precuneus activity and that this effect remained robust after controlling for Euclidean distance. Likewise, we showed that precuneus activity was positively correlated with traveled Euclidean distance during implicit navigation in the abstract social space. Moreover, the present peak coordinates coincide with those reported in a previous study by *Park et al., 2020*. In their study, participants learned 2D social hierarchy (competence and popularity) of two groups of agents and were asked to perform a transitive inference task in which they compared a pair of agents on their competence or popularity. The authors found that bilateral precuneus represented pairwise difference in the task-relevant dimension instead of pairwise Euclidean distance in the 2D space. Another study reported that posterior cingulate cortex/precuneus tracks the length of the vector of social relationship change, that is, an egocentric distance (*Tavares et al., 2015*). Taken together, we speculate that precuneus may encode the distance of the route, either on a concrete or an abstract map, that participants trespass under task demand. Less clear is the negative correlation we observed between traveled distance and activity in fusiform as well as middle occipital gyrus. Both FFG and MOG are involved in human spatial navigation (*Boccia et al., 2014*). In particular, fusiform gyrus extending into the parahippocampal place area has been shown to be involved in processing spatial information such as landmark and orientation cues (*Qiu et al., 2019*). However, to our knowledge, there is no study that reports such negative distance coding in these occipitotemporal regions during navigation.

Apart from the distance code, another focus of the current study was to test whether grid-like codes support the organization of social knowledge. Previous studies on reporting grid-like codes mostly report hexagonal modulated pattern in the EC and medial prefrontal regions aligned to putative grid orientation from both regions (*Constantinescu et al., 2016*). The current study partly replicated these findings, most of which are in the prefrontal region. In terms of prefrontal orientation, whole-brain analysis revealed alignment effects in the bilateral vmPFC and left EC. In terms of entorhinal orientation, however, alignment effect was found in the bilateral vmPFC and right superior temporal pole, but not in the EC. Though we found evidence of prefrontal grid-like codes subserving social navigation, its origin remains elusive. To date, there has been no significant evidence of grid-like neuronal tuning in this area (*Wikenheiser et al., 2021*). *Jacobs et al., 2013* examined grid-like cells in a variety of brain regions, including the frontal cortex in neurosurgical patients during a virtual navigation task, but the proportion of significant grid-like cells in the frontal cortex did not exceed type I error rate (they did identify grid-like cells in cingulate cortex, which is posterior to the prefrontal regions). Therefore, it is less likely that the prefrontal grid-like codes reported in our study and previous literature come from grid cells in the PFC. A more plausible explanation is that the anatomically and functionally connected medial- and orbitofrontal cortex and EC (*Navarro Schröder et al., 2015*; *Peng et al., 2018*; *Squire and Zola, 1996*) coordinated together to support flexible decision-making. However, adopting both univariate and multivariate approaches, the current study did not find the reliable effect of hexagonal grid-like code in the entorhinal region aligned to its own grid orientation, which should be evident if such hypothesis stands. Previous studies postulated that the lack of univariate evidence of grid-like code may have to do with a low signal-to-noise ratio (tSNR) in data from the EC (*Bao et al., 2019*). Indeed, analysis ('Materials and methods') revealed lower tSNR in all four entorhinal subregions compared to the frontal pole cluster from the quadrature filter analysis (*Figure 3—figure supplement 5A*). But we did not identify a significant relationship between tSNR and the $Z$-transformed $F$ statistics of hexagonal modulation either across participants or within

**Table 4.** Related to *Figure 3—figure supplement 5C*.
Group-level Wilcoxon signed-rank test of correlation between voxel-wise temporal signal-to-noise ratio (tSNR) and the *Z*-statistics of hexagonal modulation effect in frontal pole and entorhinal regions of interest (ROIs).

| ROIs | N | Test statistics | p |
|---|---|---|---|
| Right frontal pole | 38 | 268.000 | 0.140 |
| Left anterior–lateral entorhinal cortex | 38 | 331.000 | 0.572 |
| Right anterior–lateral entorhinal cortex | 38 | 415.000 | 0.528 |
| Left posterior–medial entorhinal cortex | 38 | 396.500 | 0.712 |
| Right posterior–medial entorhinal cortex | 38 | 433.000 | 0.369 |

participants (*Figure 3—figure supplement 5B and C*, *Table 4*). Thus, we cannot conclude that a low signal-to-noise ratio in EC led to this null finding in our study.

Nonetheless, our study did provide some incomplete evidence of grid-like activity in the EC and prefrontal region while participants were mentally trespassing an abstract social space. Grid codes are proposed to provide a metric of space (*Moser and Moser, 2008*) and underlie the process of path integration (*McNaughton et al., 2006*). Previous studies have suggested that entorhinal grid codes are capable of representing space even at a higher level of abstraction that goes beyond spatial navigation to support cognitive flexibility in general (*Bellmund et al., 2018*). Existing literature revealed correlation between entorhinal and prefrontal grid-like code and performance in spatial, perceptual, and conceptual space. Indeed, our exploratory analysis showed that the univariate hexagonal consistency effect in temporal lobe scaled with the distance effect in a comparison task block when participants chose their preferred partner for future cooperation. The distance effect reflected the extent to which participants relied on their preference judgment on the pairwise distance between the given pair of avatars. As we explicitly required participants to take both dimensions into consideration, it reflected participants' ability to plan a route between landmarks (which represents the retained impression of social encounters) in the abstract social space. This is very similar to what *Park et al., 2021* reported in their recent study. In this study, participants were trained on the rankings of 16 individuals on two dimensions (competence and popularity) separately, then they were asked to make comparisons between novel pairings unseen during training. Replicating their previous findings (*Park et al., 2020*), the authors found that the entorhinal and hippocampus system successfully reconstruct the unseen whole spectrum of social hierarchies as a 2D cognitive map. Moreover, they found that grid-like representations support trajectories of novel inferences on this 2D cognitive map that underpins decision-making. Park's study and ours differed in that they focused on discrete relational structures whilst we constructed a continuous social space. Collectively, this evidence supports the notion that grid-like codes underpin navigation in an abstract social cognitive map, be it discrete or continuous.

Intriguingly, our study found that the intensity of hexagonal modulation in the precuneus aligned to estimated grid orientation of the prefrontal region is negatively correlated with participants' social avoidance tendency. While previous studies mainly reported grid-like codes in the EC and PFC and its relevance to spatial navigation deficit in a clinically risky population (*Kunz et al., 2015*), to the best of our knowledge, no report has shown the psychological relevance of grid-like code outside these classical regions to potential social deficit. One study of direct neuronal recording revealed grid-like neuronal activity in patients' cingulate cortex during virtual spatial navigation task (*Jacobs et al., 2013*), implying that the human grid-cell network previously focused around the entorhinal and prefrontal regions also extends to the cingulate cortex. However, our study did not find evidence of grid-like pattern in the precuneus. Thus, we cannot conclude that our result points to a direct relationship between precuneus grid-like code and social functionality. Meanwhile, precuneus, as a core node of the DMN, has abundant neuroanatomical connections with prefrontal regions (*Greicius et al., 2009*; *Khalsa et al., 2014*; *Oane et al., 2020*). Precuneus has also been reported to show a decrease in resting-state connectivity to parahippocampal gyrus and medial PFC in patients with SAD (*Yuan et al., 2018*). Therefore, it is likely that downstream projections from the OFC/vmPFC motivated the observed correlation pattern in our study.

In conclusion, the present study demonstrates that navigating in a continuous social space recruited distance codes and grid-like codes in brain regions reported by spatial navigation studies and their behavioral and psychological relevance. Our findings further strengthen the notion that neural mechanisms involved in spatial cognitive map may play a domain-general role in maintaining an abstract, structured representation of knowledge that supports flexible cognitive behavior.

Our study has a few limitations. First, as we used a visual analog to guide participants to imagine moving in this abstract social space, grid-like coding could reflect the processing of both the sensory information and the abstract social concept. It would be better to control for this as in *Bao et al., 2019*. Second, to make our study comparable to those investigating the representation of abstract nonsocial knowledge, our scanner task investigates social navigation in a static environment and no social interaction or social decision is involved. Further studies should be conducted to investigate how such a structured representation of social knowledge is reused and updated during social decisions and social interactions. Third, we specifically designed the abstract social value map to be formed by two orthogonal dimensions that have the same scale. This allows us to examine grid-like activity pattern with sixfold periodicity unambiguously. But this set-up may lack generality. For one thing, it is unlikely that all parts of the whole social value space span by the warmth and competence dimension are equally important. Social ties are more likely to be formed among humans alike (*McPherson et al., 2001*) and humans show the tendency to represent ingroup members more differentially than outgroup members (*Hugenberg et al., 2010*; *Ostrom et al., 1993*), so it is likely that some area of the social value space is represented in finer granularity than the other. In spatial navigation, entorhinal grid cells showed sensitivity to geometric and environmental features, and the resulting distorted grid fields no longer form a regular hexagonal lattice and fail to show sixfold periodicity (*Derdikman et al., 2009*; *He and Brown, 2019*; *Krupic et al., 2015*). For instance, grid fields have been found to sometimes warp toward reward (*Boccara et al., 2019*). In this sense, spontaneous social value space may bear more resemblance to such kind of reward map covered by distorted grid fields rather than a regular, uniform square/circular open arena where sixfold symmetric grid-like coding was originally identified. For another, the geometric properties of social and nonsocial cognitive space in real life can be very different from that of the physical space in navigation. Our knowledge of other people in real life is often high-dimensional integrating multimodal information from sensory features (face, voice, etc.) to social cognitive information (hobbies, social status, etc.). Moreover, while dimensions of physical space (the cartesian axes) are orthogonal and their metrics are on the same scale, the dimensions of social perception in real life may not be orthogonal and may be incomparable. The stereotype content model predicted that there is a weak but low correlation between warmth and competence (*Cuddy et al., 2009*) and other studies have found there exists compensation between these dimensions (*Kervyn et al., 2010*). To the best of our knowledge, it remains elusive whether and how grid cells can represent such high-dimensional space with a non-orthonormal basis and how the univariate and multivariate analyses pipeline used to identify grid-like coding in fMRI signal could be revised to address these concerns. Taken together, the intrinsic social cognitive map in which humans structure their knowledge of others may be nonuniform, high-dimensional, and have non-orthonormal dimensions. Future studies would need to take this into account and explore beyond the sixfold grid-like activity pattern in regular 2D space.

# Materials and methods
## Participants
No power analysis was done to predetermine the sample size. Instead, we tried to achieve a sample size comparable to previous studies investigating grid-like representations using fMRI (*Doeller et al., 2010*). Forty-four participants (18 males, mean age ± SD: 21.59 ± 2.56) recruited from surrounding universities finished all behavioral training and fMRI scanning in the current study at the Beijing Normal University Imaging Center for Brain Research. Each participant underwent intensive behavioral training (including one learning session and one review session) and one fMRI scanning session (*Figure 1—figure supplement 1F*). Three participants were excluded from all further analyses due to excessive head motion in the scanner (framewise displacement >3 mm). Another three participants failing to achieve satisfactory behavioral accuracies (<60%) in the scanning session were excluded

from all further analyses. The remaining 38 participants' data were included in the behavioral and fMRI analysis reported in the main text (15 males, mean age ± SD: 21.47 ± 2.64).

The study was approved by the ethics committee of the National Key Laboratory of Cognitive Neuroscience and Learning at Beijing Normal University (ICBIR_A_0071_011). All regulations were followed, and participants signed paper-form informed consent before the experiment. Participants received monetary compensation for their participation in the current study.

## Data acquisition

### Behavioral data acquisition

Behavioral tasks were programmed using Psychophysics Toolbox-3 in MATLAB or E-prime 2.0. Online pilot rating study, pre- and post-experiment questionnaires were collected via Qualtrics.

### Pre-rating of experiment stimuli

Avatars were represented by standard photographs of volunteers who consented to the use of their photographs in the experiment. To make sure the photographs did not induce prior perception of competence and trustworthiness, we conducted a pilot rating experiment online via Qualtrics in an independent sample of 39 participants (17 males, age: 22.56 ± 2.41). In each trial, participants were shown one face and were asked to indicate their rating of the face on a 9-point Likert scale. Rating items were selected to reflect participants' perception of each face's competence, trustworthiness, and attractiveness based solely on first impressions. Items reflecting competence included ability, efficacy, and creativity. Items reflecting trustworthiness included morality, friendliness, trustworthiness, and sincerity. Attractiveness was also included as an additional rating item. The order of rating items was randomized across trials to avoid habitual responses. We selected 6 (three from each gender) out of 48 faces that received middling ratings on all three dimensions (competence, trustworthiness, plus attractiveness) to serve as the stimuli in the current study. One-sample *t*-tests revealed that ratings on all three dimensions of these six faces were not significantly different from five (the midpoint of the rating scale).

### Investment task

At the beginning of the experiment, participants completed an investment task to develop a concrete understanding of the quantitative meaning of the two dimensions of the social value map (*Figure 1—figure supplement 1A*). The task is modified from the classic trust game. The investor is endowed with 1000 points and must allocate them between two agents. Competence is operationalized as how much more an agent can multiply his/her received investment while the return rate signals trustworthiness. The expectation of profit (the number of points) an investor can earn from one agent is thus formulated by:

$$E\left[Profit\right] = investment \times \left(1 + competence\right) \times trustworthiness \tag{1}$$

Participants were all assigned the role of an investor. The two agents' competence and trustworthiness levels were preset by the experiment program and remained the same across participants.

### Match task

After the investment task, participants completed 30 trials of the match task to learn to use a nonspatial controller to morph the visualization (*Figure 1—figure supplement 1B*). One box with two bars appeared on the left side of the screen, and the participants were asked to use the controller to morph it to match the target box on the right side of the screen. The two boxes varied in the height of their enclosed bars, that is, the competence and trustworthiness dimensions. The controller consisted of two horizontal thick black chunks, which signified how much the respective dimension would change. The closer to the top boundary, the more that dimension would change and vice versa. If placed on the midline, then the corresponding dimension would not change. Together, the controller's two chunks represented the ratio between how much the heights of two bars (i.e., values on two dimensions) changed relative to each other. To encourage participants to integrate two dimensions simultaneously, we instructed them to morph the bars by making as few transitions as possible and as accurate as possible. The morphing was continuous, with bars in the left box shrinking or stretching in height.

### Explore task

After participants learned how to use the controller, they explored the vast social space by morphing the bars with different competence: trustworthiness ratio using the nonspatial controller to look for the six avatars (*Figure 1—figure supplement 1C*). No prior information about the characteristics of the avatars was provided to the participants. Thus, participants could only explore social space as if they were looking for landmarks in a newly introduced physical environment. An avatar would pop out when the visualization matched his/her characteristics. Specifically, the avatar would pop out when participants' current location fell within a 0.01-unit radius of the correct location on the social value map. In this way, participants not only learned the characteristics of each avatar but also became familiar with the whole space even though this map-like structure was never revealed to them.

### Collect task

After participants were fully acquainted with the avatars and the social space, they completed the collect task (*Figure 1—figure supplement 1D*). The task resembles the match task, except that instead of a target box, a target avatar appeared on the right side of the screen. Participants were instructed to morph the box on the left to match the characteristic of the avatar. Again, we instructed them to morph the bars by making as few transitions as possible and as accurate as possible. Each avatar was tested five times in each block, yielding 30 trials. Trials were presented randomly. Participants completed one block of collect task in each session.

### Recall task

Participants completed the recall task both outside and inside the scanner (*Figure 1D*, *Figure 1—animation 1*). On each trial of the recall task, participants were first shown a visualization morphing according to a predefined competence: trustworthiness ratio for 1 s. The bars then stopped morphing, and participants were instructed to imagine the bars keep morphing according to the same ratio, at the same speed, and for the same amount of time. After this, participants had to choose which of the three given options matched the bars after imagination.

Each block has 80 trials, each defined by a trajectory. To make sure trajectory directions were sampled uniformly on $[0,2\pi]$, we divided the whole range into 80 bins and each trial sampled a direction from one bin. Half of the trajectories led to learned avatars while the other half led to locations not associated with any avatar in the abstract space. Outside the scanner, participants completed two blocks of recall task in each session, resulting in 160 trials in total. In the scanner, participants completed four blocks of recall task, one in each run, resulting in 320 trials in total.

### Rank task

We asked participants to rank the six learned avatars based on competence, trustworthiness, and willingness to cooperate with the avatar. When asking participants to rank based on willingness to cooperate, we specifically asked them to consider the two dimensions simultaneously and with the same weight.

### Compare task

The compare task was designed to test whether participants formed an internal representation of the social value map, even though the map-like structure was never directly revealed to them during the experiment (*Figure 1—figure supplement 1E*). Participants were asked to compare two avatars on a given dimension and indicate which face had a higher value on the respective dimension using a key press. The hypothesis was that if internal representation were formed, then it would take participants longer time to compare avatars with closer distance on the given dimension than to compare avatars far apart. Each possible combination of avatars was tested four times on each of the two dimensions. This yields 120 trials presented randomly. After that, an additional block on willingness to cooperate with 60 trials (four repetitions per combination) followed. Again, we specifically asked participants to give the two dimensions equal consideration when deciding their willingness to cooperate.

### Pre- and post-experiment face rating

To reassure that the used face stimuli did not elicit biased social perception in the current sample, we asked our participants to rate the six avatars on a 9-point Likert scale when they signed up for

the experiment (at least 1 day before the experiment). The rating items were identical to those in the online rating pilot experiment. To test whether participants learned each avatar's characteristics and updated their perception of the avatars, we asked our participants to rate the six avatars again after the fMRI scan.

### Map task

At the end of the post-experiment questionnaire, participants were informed about the map-like structure and were asked to indicate the location of each avatar on an empty social value map using a mouse click. We instructed them that the locations of avatars were defined by his/her level of competence and trustworthiness. We also asked participants whether they were aware of such a map-like structure and whether their strategy resembled this map-like organization.

### MRI data acquisition

We acquired T2-weighted functional images on a 3T SIEMENS MAGNETOM Prisma scanner with a 64-channel head coil. We acquired 33 slices, 3 mm thick with repetition time (TR) = 2000 ms, echo time (TE) = 30 ms, flip angle = 90°, field of view (FoV) = 224 mm, voxel size = 3.5 × 3.5 × 3.5 mm$^3$. To correct for spatial distortion, a field map was acquired with dual echo-time images covering the whole brain with the following parameters: TR = 400 ms, TE1 = 4.92 ms, TE2 = 7.38 ms, flip angle = 60°, FoV = 224 mm, voxel size = 3.5 × 3.5 × 3.5 mm$^3$. A T1-weighted structural image was acquired with the following parameters: TR = 2530 ms, TE = 2.98 ms, flip angle = 7°, FoV = 256 mm, voxel size = 0.5 × 0.5 ×1 mm$^3$.

## Data analysis

All analyses were performed with custom code in MATLAB and R.

### Behavioral data analysis

Different performance indices were calculated for each task to test whether performance improved over sessions. To test whether performance improved over sessions, we built linear mixed effect models with random intercept for each participant and entered session as fixed effect otherwise specified.

In the explore task, we first computed the time participants spent exploring the social space to find all six avatars. We predicted that participants would spend much less time in the review session than the learning session had they been fully acquainted with the space and the avatars. Next, the social space was divided into 15 × 15 subregions and we computed the amount of time spent in each subregion and plotted the corresponding color-coded trajectory maps (*Figure 1E*). The top/bottom rows and leftmost/rightmost columns of subregions were classified as edges, yielding the index of 'time at edges'. In addition, 'time at avatars' was computed based on the time the avatars were on screen. Note that all these timing indices were computed as a percentage of the total time spent navigating in the explore task in a given session. If participants were well acquainted with the space and avatars, 'time at edges' would decrease and 'time at avatars' would increase in the review session compared to the learning session. We built linear mixed effect models to examine these hypotheses.

In the collect task, we computed the following metrics:

1. The mean number of transitions participants needed to morph the visualization to match the target avatar.
2. the deviation from ideal trajectory, which is the angle difference between participants' first transition and the ideal trajectory in each trial. We computed the mean deviation and the percentage of trials where participants deviated less than 15°.
3. The mean distance between the target avatar and participants' response.

Greater memory performance should be signaled by a lower mean number of transitions, a higher percentage of trials with deviation less than 15°, and a shorter mean distance from the target avatar.

In the recall task, we computed the percentage of correct responses.

In the compare task, we focused on the effect of distance between compared pairs on accuracy and response time. We defined task-relevant distance as the distance between avatar pairs on the compared dimension. Specifically, for the two social value map dimensions, this was the distance on

the competence/trustworthiness axis. For the willingness to cooperate block, this was the difference between the expected profit calculated in formula 1. We concatenated data from the review session and the scanning session and built linear mixed effect models. Session and task-relevant distance between avatar pairs was entered as fixed effect. We included random intercept for each participant. We also included a random slope term for the task-relevant distance. The random slopes were extracted from the mixed effect model as indices for distance effect when exploring the behavioral relevance of spatial codes for the social value map.

For the pre- and post-experiment face ratings. we built three separate linear mixed effect models with random intercept to test whether participants' ratings on competence, trustworthiness, and attractiveness became more aligned with the avatars' social characteristic after the experiment. Three regressors were included as fixed effect: (1) time, that is, post- vs pre-experiment; (2) avatar's social characteristic; and (3) interaction between time and avatar. Specifically, avatar's social characteristic was defined in alignment with the rating item entered as the dependent variable. That is, in the competence rating regression model, social characteristic referred to the location on competence axis. Similarly, in the trustworthiness rating regression model, social characteristic referred to the location on the trustworthiness axis. Finally, in the attractiveness rating regression model, expected profit was entered as a social characteristic.

## Preprocessing of fMRI data

Preprocessing of fMRI data was done with SPM12. Functional images were spatially realigned to the first image in the time series and corrected for slice-timing. Spatial distortion was based on a field map. The T1-weighted structure image was co-registered to the mean aligned functional image, segmented, and normalized to MNI space. The derived transformation parameters from structural image normalization were applied to normalize realigned functional images. Finally, smoothing was done with a 6 mm full-width half-maximum Gaussian kernel.

## Whole-brain univariate analysis

All whole-brain univariate analyses were conducted in SPM12 following routine procedure. In particular, we switched off SPM12's implicit threshold during model estimation and used an explicit mask from SPM12's default repository (the 'mask_ICV.nii' file) that included all intracranial volumes instead. This specific treatment was done as we observed susceptible signal loss in the frontal and entorhinal regions after applying the implicit threshold. Following routines from previous literature, the explicit mask was used instead. In all univariate analyses, boxcar functions were used and the boxcar duration corresponds to the duration of the modelled stage.

### A functional localizer for hexagonal modulation: GLM1

GLM1 consisted of two regressors, one modeling the morph stage and one modeling the choice stage. The aim of GLM1 was to identify the regions sensitive to hexagonal modulation. Let $\varphi$ denote the hypothetical grid orientation of a region, and $\theta$ the trajectory (moving direction). If neural activity in a region is hexagonally modulated, then its activity should be a waveform of $cos\left[6*(\theta-\varphi)\right]$. Omitting error and intercept, this hypothesis can be expressed using the following formula:

$$Activity = \omega * cos\left[6*(\theta-\varphi)\right]$$

$$\omega * cos6\varphi cos6\theta + \omega * sin6\varphi sin6\theta \tag{2}$$

$$\beta_{cos6\theta}cos6\theta + \beta_{sin6\theta}sin6\theta \tag{3}$$

where: $\beta_{cos6\theta} = \omega * cos6\varphi$

$$\beta_{sin6\theta} = \omega * sin6\varphi$$

If the theoretical prediction that there is an effect of hexagonal modulation is true, $\omega$ should be significantly different from zero.

Based on this, a pair of sine and cosine regressors were used as parametric modulators for the morph stage. In this way, testing against the null hypothesis of $\beta_{cos6\theta} = \beta_{sin6\theta} = 0$ is essentially testing against the null hypothesis of $\omega = 0$. Therefore, at the individual level, F-test was used to search for potential regions modulated by a linear combination of $\beta_{cos6\theta}cos6\theta + \beta_{sin6\theta}sin6\theta$. These individual

*F*-statistics were transformed into *Z*-statistics before entering group-level one-sample *t*-test. In future correlation analyses, the *Z*-statistics were extracted as an index for hexagonal modulation effect.

## Iterative cross-validation analysis for hexagonal consistency: GLM2

By binning trials according to trajectory's alignment to a putative grid orientation $\varphi$, GLM2 aimed to test for hexagonal consistency.

In this cross-validation procedure, we separated each participant's data into two sets, an estimating set with three runs and a testing set with one run. The estimating set was used to calculate the grid orientation of the ROIs. The remaining one run then served as the testing set where the alignment effect was tested with the inferred grid orientation. The theoretical prediction for hexagonal consistent grid-like code is that neural signal should be stronger in trials aligned than misaligned to the grid orientation.

To estimate grid orientation, GLM1 was applied to the estimating set yielding beta estimates for the sine and cosine regressors ($\beta_{sin6\theta}$ and $\beta_{cos6\theta}$) in each voxel. We extracted these beta estimates from ROI masks. Then, within a given ROI, beta estimates were averaged across all voxels within this region for sine and cosine regressors, respectively. The averaged beta estimates ($\bar{\beta_{sin}}$ and $\bar{\beta_{cos}}$) were used to calculate the grid orientation for this region (grid orientation $\varphi = \left[ artan\left( \frac{\bar{\beta_{sin}}}{\bar{\beta_{cos}}} \right) \right] /6$).

To test for the prediction of consistency effect, we classified trials according to $\theta$'s (trajectory direction) offset from $\varphi$ into 12 bins of 30° (***Figure 3A***, left panel), yielding six bins of aligned trials (0° modulo 60°) and six bins of misaligned trials (30° modulo 60°).

The above steps yielded the grid orientation applied to classify trials in the testing set. Each run acted as an estimating set three times and received putative grid orientation as a testing set once. Eventually, the four testing sets were fitted with GLM2. GLM2 consisted of 12 regressors, each modeling the morph stage of one bin of trials. At the individual level, contrast was built to test the activity difference between aligned and misaligned trials (align > misalign). These first-level contrasts were entered into second-level one-sample *t*-test to look for the regions that show hexagonal consistency.

## Identifying distance code with parametric modulation: GLM3

GLM3 resembled GLM1, which also consisted of two regressors, one modeling the morph stage and one modeling the choice stage. The aim of GLM3 was to identify the regions sensitive to distance modulation. Thus, we entered the traveled Euclidean distance during the morph stage as a parametric modulator for the morph-stage regressor. At the individual level, *t*-contrast was built to search for potential regions modulated by the distance parametric modulator. These contrasts were entered into second-level one-sample *t*-test to look for the regions that represent distance during social navigation.

## Behavioral relevance of spatial codes

To specifically test the effect of grid-like code in regions that survive correction in GLM2 on behavior, we extracted the beta estimates of align > misalign contrast for each region and each participant. Then we computed Pearson correlation between these beta estimates and accuracy as well as response time in the recall task in the scanner. This procedure was applied to the regions demonstrating distance representation (GLM3) as well. As multiple tests were performed, we adopted false discovery rate (FDR) correction to p-values to lower FDR. No significant correlation was found (***Figure 4—figure supplement 1***).

For whole-brain analyses to explore whether there is any correlation between neural indices of map-like representation and behavior performance outside the scanner as well as individual differences, we calculated two categories of behavioral indices to explore the relevance of the spatial codes calculated above. The first category of indices reflects participants' ability to make decision based on the social value map. Specifically, we considered the distance effect during the cooperation block when participants decide the avatar they are more willing to cooperate with when taking into account both dimensions. Distance effect indices were extracted from the linear mixed effect model estimates specified in the behavioral data analysis section. The second category of indices reflects participants' social trait, including social anxiety and social avoidance scores. We explored the behavioral relevance of grid-like code by entering the above indices as covariates into the second-level analysis when testing the grid orientation consistency effect (GLM2), and all covariates were tested in separate regression models.

### Univariate and multivariate analyses in EC ROI

As we failed to find evidence of grid-like activity in EC aligned to its own putative grid orientation using the first approach, we took the anatomical masks of EC (*Maass et al., 2015*) and explored grid-like activity using both the orientation-estimation and RSA approach.

#### Univariate analyses

For the univariate analysis in ROI, we looked at both the quadrature filter analysis and the hexagonal consistency analysis. For the former, we extracted the *Z*-statistics in GLM1 from different subregions of EC for each participants and conducted one-sample *t*-test to see whether it is above zero in any of the subregions. For the hexagonal consistency effect, we extracted the contrast estimate for align > misalign in GLM2 from different subregions of EC for each participants and conducted one-sample *t*-test to see whether it is above zero in any of the subregions. We looked at both within-subregion consistency (i.e., contrast estimates extracted from the same subregion as the orientation estimation subregion) and across-subregion consistency (i.e., contrast estimates extracted from another subregion). As multiple tests were performed, we adopted FDR correction to p-values to lower FDR.

#### Multivariate analyses

The second approach leverages RSA and is widely adopted based on the assumption that a distributed coding scheme is employed by the EC. We conducted conventional representational similarity analysis on the unsmoothed data as implemented by CosmoMVPA toolbox (*Oosterhof et al., 2016*) in MATLAB.

First, we estimated trial-specific activation for each trial in each run. As trajectories were drawn randomly in the social space, each trial was defined by a unique trajectory, yielding a rich-condition design. It has been suggested that in such fast event-related (fast ER) designs, signal for nearby trials tend to overlap in time. Conventional approach in block or slow ER design that includes each trial as a separate regressor in a large model may be problematic under fast ER settings as estimates can become unstable due to collinearity between the trial-specific regressors. To obtain a more accurate estimate of trial-specific activation, we leveraged the 'least squares separate' (LSS) approach (*Abdulrahman and Henson, 2016*; *Mumford et al., 2012*). This approach built separate GLMs for each trial, with one target regressor modeling the current trial of interest and a second nuisance regressor modeling the rest. After that, we took the beta estimate of the target regressor as the estimation of trial-specific activation. Signals were extracted from the entorhinal ROIs (from entorhinal as a whole or from four subregions) as trial-specific activity pattern for each of the 320 trials (*Figure 3—figure supplement 1A*).

We then tested whether this multivariate pattern is hexagonally modulated. Though the exact implementations varied slightly across studies, they fell into the following two categories according to their dependence on the estimated putative grid orientation.

The 'orientation-independent' approach (*Figure 3—figure supplement 1B*) tested the hypothesis that multivariate pattern similarity between trajectory pairs is proportional to the angular difference between their moving direction modulated by 60° (*Bellmund et al., 2016*; *Viganò and Piazza, 2020*; *Viganò et al., 2021*). We first calculated the representational similarity between all possible pairs of trials, yielding a 320 × 320 dissimilarity matrix (DSM) for neural data (*Figure 3—figure supplement 1A*). Then, we derived model DSM from theoretical prediction (*Figure 3—figure supplement 1B*). Finally, we computed Spearman's rank correlation between neural DSM and model DSM for each participant. One-sample *t*-test was conducted across participants on the Fisher *Z*-transformed correlation coefficients to test whether the correlation was significantly above zero.

The 'orientation-dependent' approach (*Figure 3—figure supplement 1C*) leveraged the assumption that if activity in a given ROI is aligned to a preferred orientation, the similarity of multivoxel patterns in this region should be higher for aligned trial pairs than that between aligned and misaligned trial pairs (*Bao et al., 2019*). We computed the grid orientation in each entorhinal subregion using the same leave-one-out cross-validation procedure as in the univariate analysis. Trials were then classified into aligned and misaligned trials according to the angular difference between trajectories and the estimated grid orientation. We then computed the similarities between aligned-aligned trial pairs (AA) and aligned-misaligned trial pairs (AM). By subtracting the similarity between AM pairs from the similarity between AA pairs, we yield a pattern similarity difference index for each participant

(*Figure 3—figure supplement 1C*). Paired *t*-tests were conducted to test whether the differences were significantly greater than zero. As multiple tests were performed, we adopted FDR correction to p-values to lower FD.

### Testing distribution of grid orientation

All tests in this section are circular statistics (*Mardia and Jupp, 1999*) implemented by the MATLAB toolbox *CircStat* (*Berens, 2009*).

To test whether the grid orientations within an ROI are similar, we calculated the putative grid orientations for each voxel in an ROI using voxel-wise beta estimates of the sine and cosine regressors from GLM1. Then the vector of putative voxel-wise grid orientations is entered into a *V*-test to test whether the distribution of the orientations is clustered. We specifically used the *V*-test instead of Rayleigh's test because the alternative hypothesis in *V*-test assumed that data have a known mean direction. This is more in line with the hypothesis that neighboring grid cells share similar grid orientation. As the cross-validation procedure yields three estimating sets for each participant, this procedure is repeated for all estimating sets and all participants.

To test the distribution of grid orientation in the population, for one ROI, we calculated the grid orientation for each participant using the averaged beta estimates of the sine and cosine regressors from GLM1 in this ROI. This vector of grid orientation is entered into Rayleigh's test. Following the previous study, we assume that the grid orientations across participants would be uniformly distributed.

### Relationship between temporal signal-to-noise ratio (tSNR) and hexagonal modulation effect

Following previous literature (*Murphy et al., 2007*; *Triantafyllou et al., 2005*), we calculated the voxel-wise tSNR of a time series, that is, the tSNR for a single voxel in a single run of one participant, as

$$tSNR = \frac{\mu}{\sigma}$$

where μ is the mean activity of the timeseries, and $\sigma$ is its standard deviation.

The tSNR of each voxel is calculated as the mean across four runs, and the tSNR for each ROI is calculated as the averaged tSNR of all voxels in the ROI.

To test the relationship between tSNR and hexagonal modulation effect, we conducted correlation analysis across participants and within each participant. For the across-participant analysis, we tested the correlation between the mean hexagonal modulation effect in an ROI and its mean tSNR across all participants. For the within-participants analysis, for each participant, in an ROI, we calculated the correlation coefficient between voxel-wise hexagonal modulation effect and voxel-wise tSNR. Then, after Fisher *Z* transformation, we tested whether the participant-level correlation coefficients were significantly different from zero using Wilcoxon signed-rank test at group level.

## Acknowledgements

We are grateful for the volunteers who participated in this study. We thank Alexandra Constantinescu for providing helpful information regarding the experiment design in their previous study. We thank Kelou Jin for his hard work and help when conducting pilot experiments. We thank Shen Zhang and Huagen Wang for their help in data analysis. This work was supported by the Scientific and Technological Innovation (STI) 2030-Major Projects (2021ZD0200500), the National Natural Science Foundation of China (32130045 and 32271092), the Major Project of National Social Science Foundation (19ZDA363), the Beijing Municipal Science and Technology Commission (Z151100003915122), and the National Program for Support of Top-notch Young Professionals.

## Additional information

### Funding

| Funder | Grant reference number | Author |
|---|---|---|
| Ministry of Science and Technology of the People's Republic of China | 2021ZD0200500 | Chao Liu |
| National Natural Science Foundation of China | 32130045 | Shaozheng Qin |
| National Natural Science Foundation of China | 32271092 | Chao Liu |
| the Major Project of National Social Science Foundation | 19ZDA363 | Chao Liu |
| the Beijing Municipal Science and Technology Commission | Z151100003915122 | Chao Liu |
| National Program for Support of Top-notch Young Professionals | | Chao Liu |

The funders had no role in study design, data collection and interpretation, or the decision to submit the work for publication.

### Author contributions

Zilu Liang, Conceptualization, Formal analysis, Investigation, Visualization, Writing - original draft, Writing - review and editing; Simeng Wu, Jie Wu, Investigation; Wen-Xu Wang, Conceptualization; Shaozheng Qin, Chao Liu, Conceptualization, Supervision, Funding acquisition, Writing - review and editing

### Author ORCIDs

Zilu Liang (iD) http://orcid.org/0009-0009-3446-5117
Wen-Xu Wang (iD) http://orcid.org/0000-0002-4170-8676
Shaozheng Qin (iD) http://orcid.org/0000-0002-1859-2150
Chao Liu (iD) http://orcid.org/0000-0003-1149-2314

Reviewer #1 (Public Review): https://doi.org/10.7554/eLife.89025.4.sa1
Reviewer #3 (Public Review): https://doi.org/10.7554/eLife.89025.4.sa2
Author response https://doi.org/10.7554/eLife.89025.4.sa3

## Additional files

### Supplementary files

• MDAR checklist

### Data availability

We have published preprocessed smoothed and unsmoothed functional MRI data, the code to run the experiment and the code to run the univariate analysis reported in the main text. Dataset and code are publicly available at https://doi.org/10.57760/sciencedb.08637.

The following dataset was generated:

| Author(s) | Year | Dataset title | Dataset URL | Database and Identifier |
|---|---|---|---|---|
| Liang Z, Wu S, Wu J, Wang W, Qin S, Liu C | 2023 | Social navigation: distance and grid-like codes support navigation of abstract social space in human brain | https://doi.org/10.57760/sciencedb.08637 | Science Data Bank, 10.57760/sciencedb.08637 |

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
