## [Editor Report · eLife assessment]

This study tackles a significant question: Does the brain apply spatial navigation systems to evaluate decision options in conceptual social spaces? The investigation is **useful** as it seeks to address this intriguing hypothesis. The findings offer partial support: a **solid** analysis revealed characteristic grid-like patterns associated with decision-making directions. However, it remains uncertain whether these effects are genuinely due to navigating a conceptual social space or potentially confounded by changes in visual stimuli. The experimental design may not be capable of definitively resolving this issue.

---

## [Referee Report · Reviewer #1 (Public Review)]

The study offers intriguing insights, yet interpretations warrant caution, as the authors themselves acknowledged in their discussion of limitations.

The observed grid-like neural activity might not signify navigating a social landscape but rather a sensory feature space. The study's design had participants associate each face with a pair of bar lengths, with the purported 'navigation' being merely a response to the morphing of bar graph images. Crucially, the task did not necessitate any social cognitive processing to estimate grid-like activity. When making social decisions in a separate task, it is unclear whether participants were actually traversing a social space mentally or simply recalling the bar graphs linked to each face to calculate decision values. Notably, during the trust game, competence and trustworthiness did not equally influence decision-making (as illustrated by Equation 1), implying the possibility that the space represented may be more perceptual than social in nature.

The existence of a universal brain representation for faces within a social context is still debatable. Participants were not required to form a cognitive map of the six faces based on social traits; they could simply remember each face's trait values. While the study suggests that reaction times correlated with the perceived social distances between faces hint at the creation of internal representations, this phenomenon could occur without a true cognitive map of the face relationships. To convincingly argue for such internal representations in the brain, additional multivariate pattern analysis would be necessary to demonstrate that these are not merely the result of perceptual differences in the bar graphs associated with each face.

---

## [Referee Report · Reviewer #3 (Public Review)]

Liang and colleagues set out to test whether the human brain uses distance and grid-like codes in social knowledge using a design where participants had to navigate in a two-dimensional social space based on competence and warmth during an fMRI scan. They showed that participants were able to navigate the social space and found distance-based codes as well as grid-like codes in various brain regions, and the grid-like code correlated with behavior (reaction times).

On the whole, the experiment is designed appropriately for testing for distant-based and grid-like codes, and is relatively well powered for this type of study, with a large amount of behavioral training per participant. They revealed that a number of brain regions correlated positively or negatively with distance in the social space, and found grid-like codes in the frontal polar cortex and posterior medial entorhinal cortex, the latter in line with prior findings on grid-like activity in entorhinal cortex. The current paper seems quite similar conceptually and in design to previous work, most notably Park et al., 2021, Nature Neuroscience.

(1) The authors claim that this study provides evidence that humans use a spatial / grid code for abstract knowledge like social knowledge.

This data does specifically not add anything new to this argument. As with almost all studies that test for a grid code in a similar "conceptual" space (not only the current study), the problem is that, when the space is not a uniform, square/circular space, and 2-dimensional then there is no reason the code will be perfectly grid like, i.e., show six-fold symmetry. In real world scenarios of social space (as well as navigation, semantic concepts), it must be higher dimensional - or at least more than two dimensional. It is unclear if this generalizes to larger spaces where not all part of the space is relevant. Modelling work from Tim Behrens' lab (e.g., Whittington et al., 2020) and Bradley Love's lab (e.g., Mok & Love, 2019) have shown/argued this to be the case. In experimental work, like in mazes from the Mosers' labs (e.g., Derdikman et al., 2009), or trapezoid environments from the O'Keefe lab (Krupic et al., 2015), there are distortions in mEC cells, and would not pass as grid cells in terms of the six-fold symmetry criterion.

After revision, the authors now discuss some of this and the limitations and notes that future work is required to address the problem.

---

## [Author Response]

The following is the authors’ response to the previous reviews.

We would like to first thank the Editor as well as the three reviewers for their enthusiasm and conducting another careful evaluation of our manuscript. We appreciate their thoughtful and constructive comments and suggestions. Some concerns regarding experimental design, data analysis, and over-interpretation of our findings still remains unresolved after the initial revision. Here we endeavored to address these remaining concerns through further refinement of our writing, and inclusion of these concerns in the discussion session. We hope our response can better explain the rationale of our experimental design and data interpretation. In addition, we also acknowledge the limitations of our present study, so that it will benefit future investigations into this topic. Our detail responses are provided below.

**Reviewer #1 (Public Review):**
This study examines whether the human brain uses a hexagonal grid-like representation to navigate in a non-spatial space constructed by competence and trustworthiness. To test this, the authors asked human participants to learn the levels of competence and trustworthiness for six faces by associating them with specific lengths of bar graphs that indicate their levels in each trait. After learning, participants were asked to extrapolate the location from the partially observed morphing bar graphs. Using fMRI, the authors identified brain areas where activity is modulated by the angles of morphing trajectories in six-fold symmetry. The strength of this paper lies in the question it attempts to address. Specifically, the question of whether and how the human brain uses grid-like representations not only for spatial navigation but also for navigating abstract concepts, such as social space, and guiding everyday decision-making. This question is of emerging importance.I acknowledge the authors' efforts to address the comments received. However, my concerns persist:

Thanks very much again for the re-evaluation and comments. Please find our revision plans to each comment below.

(1) The authors contend that shorter reaction times correlated with increased distances between individuals in social space imply that participants construct and utilize two-dimensional representations. This method is adapted from a previous study by Park et al. Yet, there is a fundamental distinction between the two studies. In the prior work, participants learned relationships between adjacent individuals, receiving feedback on their decisions, akin to learning spatial locations during navigation. This setup leads to two different predictions: If participants rely on memory to infer relationships, recalling more pairs would be necessary for distant individuals than for closer ones. Conversely, if participants can directly gauge distances using a cognitive map, they would estimate distances between far individuals as quickly as for closer ones. Consequently, as the authors suggest, reaction times ought to decrease with increasing decision value, which, in this context, corresponds to distances. However, the current study allowed participants to compare all possible pairs without restricting learning experiences, rendering the application of the same methodology for testing two-dimensional representations inappropriate. In this study, the results could be interpreted as participants not forming and utilizing two-dimensional representations.

We apologize for not being clear enough about our task design, we have made relevant changes in the methodology section in the manuscript to make it clearer. The reviewer’s concern is that participants learned about all the pairs in the comparison task which makes the distance effect invalid. We would like to clarify that during all the memory test tasks (the comparison task, the collect task and the recall task outside and inside scanner), participants never received feedback on whether their responses were correct or not. Therefore, the comparison task in our study is similar to the previous study by Park et al. (2021). Participants do not have access to correct responses for all possible pairs of comparison prior to or during this task, they would need to make inference based on memory retrieval.

(2) The confounding of visual features with the value of social decision-making complicates the interpretation of this study's results. It remains unclear whether the observed grid-like effects are due to visual features or are genuinely indicative of value-based decision-making, as argued by the authors. Contrary to the authors' argument, this issue was not present in the previous study (Constantinescu et al.). In that study, participants associated specific stimuli with the identities of hidden items, but these stimuli were not linked to decision-making values (i.e., no image was considered superior to another). The current study's paradigm is more akin to that of Bao et al., which the authors mention in the context of RSA analysis. Indeed, Bao et al. controlled the length of the bars specifically to address the problem highlighted here. Regrettably, in the current paradigm, this conflation remains inseparable.

We’d like to thank the reviewer for facilitating the discussion on the question of ‘social space’ vs. ‘sensory space’. The task in scanner did not require value-based decision making. It is akin to both the Bao et al. (2019) study and Constantinescu et al. (2016) study in a sense that all three tasks are trying to ask participants to imagine moving along a trajectory in an abstract, non-physical space and the trajectory is grounded in sensory cue. Participants were trained to associate the sensory cue with abstract (social/nonsocial) concepts. We think that the paradigm is a relatively faithful replication of the study by Constantinescu et al. Nonetheless, we agreed that a design similar to Bao et al. (2019) which controls for sensory confounds would be more ideal to address this concern, or adopting a value-based decision-making task in the scanner similar to that by Park et al. (2021), and we have included this limitation in the discussion section.

(3) While the authors have responded to comments in the public review, my concerns noted in the Recommendation section remain unaddressed. As indicated in my recommendations, there are aspects of the authors' methodology and results that I find difficult to comprehend. Resolving these issues is imperative to facilitate an appropriate review in subsequent stages.Considering that the issues raised in the previous comments remain unresolved, I have retained my earlier comments below for review.

We apologize for not addressing the recommendations properly, please find detailed our response and plans for revision.

I have some comments. I hope that these can help.(1) While the explanation of Fig.4A-C is lacking in both the main text and figure legend, I am not sure if I understand this finding correctly. Did the authors find the effects of hexagonal modulation in the medial temporal gyrus and lingual gyrus correlate with the individual differences in the extent to which their reaction times were associated with the distances between faces when choosing a better collaborator? If so, I am not sure what argument the authors try to draw from these findings. Do the authors argue that these brain areas show hexagonal modulation, which was not supported in the previous analysis (Fig.3)? What is the level of correlation between these behavioral measures and the grid consistency effects in the vmPFC and EC, where the authors found actual grid-like activity? How do the authors interpret this finding? More importantly, how does this finding associate with other findings and the argument of the study?

We apologize for not being clear enough in the manuscript and we will improve the clarity in our revision. This exploratory analysis reported in Figure 4 aims to use whole-brain analysis to examine: (1) if there is any correlation between the strength of grid-like representation of social value map and behavioral indicators of map-like representation; and (2) if there are any correlation between the strength of grid-like representation of this social value map and participants’ social trait.

To be more specific, for the behavioral indicator, we used the distance effect in the reaction time of the comparison task outside the scanner. We interpreted stronger distance effect as a behavioral index of having better internal map-like representation. We interpreted stronger grid consistency effect as a neural index of better representation of the 2D social space. Therefore, we’d like to see if there exists correlation between behavioral and neural indices of map-like representation.

To achieve this goal, behavioral indicators are entered as covariates in second-level analysis of the GLM testing grid consistency effect (GLM2). Figure3 showed results from GLM2 without the covariates. Figure4 showed results of clusters whose neural indices of map-like representation covaried with that from behavior and survived multiple-comparison correction. Indeed, in these regions, the grid consistency effect was not significant at group level (so not shown in Figure 3). We tried to interpret this finding in our discussion (line 374-289 for temporal lobe correlation, line 395-404 for precuneus correlation).

Finally, we would like to point out that including the covariates in GLM2 did not change results in Figure3, the clusters in Figure3 still survives correction. Meanwhile, these clusters in Figure 3 did not show correlation with behavioral indicators of map-like representation.

(2) There are no behavioral results provided. How accurately did participants perform each of the tasks? How are the effects of grid consistency associated with the level of accuracy in the map test?Why did participants perform the recall task again outside the scanner?

We will endeavor to improve signposting the corresponding figures in the main text. For the behavioral results, we reported the stats in section “Participants construct social value map after associative learning of avatars and corresponding characteristics” in the main text, and the plots are shown in Figure 1. Particularly, figure 1F showed accuracy of tasks in training, as well as the recall task in the scanner. For the correlation, we did not find significant correlation between behavioural accuracy and grid consistency effect. We will make it clearer in the result section.

(3) The methods did not explain how the grid orientation was estimated and what the regressors were in GLM2. I don't think equations 2 and 3 are quite right.

For the grid orientation estimation method, we provided detailed description in the Supplementary methods 2.2.2. We will add links to this section in the main text.

Equation 2 and 3 describes how the parametric regressors entered into GLM2 were formed and provided prerequisites on calculation of grid orientations. Equation 2 was the results of directly applying the angle addition and subtraction theorems so they should be correct. We will try to make the rationale clearer in the supplementary text.

(4) With the increase in navigation distances, more grid cells would activate. Therefore, in theory, the activity in the entorhinal cortex should increase with the Euclidean distances, which has not been found here. I wonder if there was enough variability in the Euclidean distances that can be captured by neural correlates. This would require including the distributions of Euclidean distances according to their trajectory angles. Regarding how Fig.1E is generated, I don't understand what this heat map indicates. Additionally, it needs to be confirmed if the grid effects remain while controlling for the Euclidean distances of navigation trajectories.

We did not specifically control for the trajectory length, we only controlled for the distribution of trajectory to be uniform. We have included a figure of the distribution of Euclidean distances in Figure S9 and the distribution of trajectory direction in Figure S8.

**Author response image 2. sa3fig2:** 

As for Figure 1E, we aim to reproduce the findings from Figure 1F in Constantinescu et al. (2016) where they showed that participants progressively refined the locations of the outcomes through training. We divided the space into 15×15 subregions and computed the amount of time spent in each subregion and plotted Figure 1E. Brighter color in Figure 1E indicate greater amount of time spent in the corresponding subregion. Note that all these timing indices were computed as a percentage of the total time spent in the explore task in a given session. If participants were well-acquainted with the space and avatars, they would spend more time at the avatar (brighter color in avatar locations) in the review session compared to the learning session.

As for the effect of distances on grid-like representation, we did not include the distance as a parametric modulator in grid consistency effect GLM (GLM2) due to insufficient trials in each bin (6-8 trials). But there is side evidence that could potentially rule out this confound. In the distance representation analysis, we did not find distance representation in any of the clusters that have significant grid-like representation (regions in Figure 2).

**Reviewer #2 (Public Review):**
Summary:In this work, Liang et al. investigate whether an abstract social space is neurally represented by agrid-like code. They trained participants to 'navigate' around a two-dimensional space of social agents characterized by the traits warmth and competence, then measured neural activity as participants imagined navigating through this space. The primary neural analysis consisted of three procedures: (1) identifying brain regions exhibiting the hexagonal modulation characteristic of a grid-like code, (2) estimating the orientation of each region's grid, and (3) testing whether the strength of the univariate neural signal increases when a participant is navigating in a direction aligned with the grid, compared to a direction that is misaligned with the grid. From these analyses, the authors find the clearest evidence of a grid-like code in the prefrontal cortex and weaker evidence in the entorhinal cortex.Strengths:The work demonstrates the existence of a grid-like neural code for a socially-relevant task, providing evidence that such coding schemes may be relevant for a variety of two-dimensional task spaces.Weaknesses:In the revised manuscript, the authors soften their claims about finding a grid code in the entorhinal cortex and provide additional caveats about limitations in their findings. It seems that the authors and reviewers are in agreement about the following weaknesses, which were part of my original review: Claims about a grid code in the entorhinal cortex are not well-supported by the analyses presented. The whole-brain analysis does not suggest that the entorhinal cortex exhibits hexagonal modulation; the strength of the entorhinal BOLD signal does not track the putative alignment of the grid code there; multivariate analyses do not reveal any evidence of a grid-like representational geometry.In the authors' response to reviews, they provide additional clarification about their exploratory analyses examining whether behavior (i.e., reaction times) and individual difference measures (i.e., social anxiety and avoidance) can be predicted by the hexagonal modulation strength in some region X, conditional on region X having a similar estimated grid alignment with some other region Y. My guess is that readers would find it useful if some of this language were included in the main text, especially with regard to an explanation regarding the rationale for these exploratory studies.

Thank you very much again for your careful re-evaluation and suggestions. We have tried to improve our writing and incorporate the suggestions in the new revision.

**Reviewer #3 (Public Review):**
Liang and colleagues set out to test whether the human brain uses distance and grid-like codes in social knowledge using a design where participants had to navigate in a two-dimensional social space based on competence and warmth during an fMRI scan. They showed that participants were able to navigate the social space and found distance-based codes as well as grid-like codes in various brain regions, and the grid-like code correlated with behavior (reaction times).On the whole, the experiment is designed appropriately for testing for distant-based and grid-like codes, and is relatively well powered for this type of study, with a large amount of behavioral training per participant. They revealed that a number of brain regions correlated positively or negatively with distance in the social space, and found grid-like codes in the frontal polar cortex and posterior medial entorhinal cortex, the latter in line with prior findings on grid-like activity in entorhinal cortex. The current paper seems quite similar conceptually and in design to previous work, most notably Park et al., 2021, Nature Neuroscience.(1) The authors claim that this study provides evidence that humans use a spatial / grid code for abstract knowledge like social knowledge.This data does specifically not add anything new to this argument. As with almost all studies that test for a grid code in a similar "conceptual" space (not only the current study), the problem is that, when the space is not a uniform, square/circular space, and 2-dimensional then there is no reason the code will be perfectly grid like, i.e., show six-fold symmetry. In real world scenarios of social space (as well as navigation, semantic concepts), it must be higher dimensional - or at least more than two dimensional. It is unclear if this generalizes to larger spaces where not all part of the space is relevant. Modelling work from Tim Behrens' lab (e.g., Whittington et al., 2020) and Bradley Love's lab (e.g., Mok & Love, 2019) have shown/argued this to be the case. In experimental work, like in mazes from the Mosers' labs (e.g., Derdikman et al., 2009), or trapezoid environments from the O'Keefe lab (Krupic et al., 2015), there are distortions in mEC cells, and would not pass as grid cells in terms of the six-fold symmetry criterion.The authors briefly discuss the limitations of this at the very end but do not really say how this speaks to the goal of their study and the claim that social space or knowledge is organized as a grid code and if it is in fact used in the brain in their study and beyond. This issue deserves to be discussed in more depth, possibly referring to prior work that addressed this, and raise the issue for future work to address the problem - or if the authors think it is a problem at all.

Thanks very much again for your careful re-evaluation and comments. We have tried to incorporate some of the suggested papers into our discussion. In summary, we agree that there is more to six-fold symmetric code that can be utilized to represent “conceptual space”. We think that the next step for a stronger claim would be to find the representation of more spontaneous non-spatial maps.

References

Bao, X., Gjorgieva, E., Shanahan, L. K., Howard, J. D., Kahnt, T., & Gottfried, J. A. (2019). Grid-like Neural Representations Support Olfactory Navigation of a Two-Dimensional Odor Space. Neuron, 102(5), 1066-1075 e1065. https://doi.org/10.1016/j.neuron.2019.03.034

Constantinescu, A. O., O'Reilly, J. X., & Behrens, T. E. J. (2016). Organizing conceptual knowledge in humans with a gridlike code. Science, 352(6292), 1464-1468. https://doi.org/10.1126/science.aaf0941

Park, S. A., Miller, D. S., & Boorman, E. D. (2021). Inferences on a multidimensional social hierarchy use a grid-like code. Nat Neurosci, 24(9), 1292-1301. https://doi.org/10.1038/s41593-02100916-3